# Public Perception of Haze Weather Based on Weibo Comments

**DOI:** 10.3390/ijerph16234767

**Published:** 2019-11-28

**Authors:** Qiang Zhang, Jinshou Chen, Xueyan Liu

**Affiliations:** 1College of Computer Science & Engineering, Northwest Normal University, Lanzhou 730070, China; 2College of Mathematics & Statistics, Northwest Normal University, Lanzhou 730070, China; liuxy@nwnu.edu.cn

**Keywords:** social public environment perception, data mining, semantic analysis, complex network

## Abstract

In China, haze weather has become a major public concern and is frantically discussed by the public. Many people express their views, opinions, or complaints on social media. Effectively extracting this useful information may help to improve our understanding of how the public perceive and respond to haze, and could potentially contribute to environmental policy-making. In this paper, we study how the public perceive haze during haze weather and how this perception changes with seasons based on comment data from a major social media platform in China, Weibo, and using several social network methods, including perceptual keyword cloud mapping, complex network topology characteristics, and social perception analysis. The results showed that the public’s perception was focused on the causes of haze in spring, enjoyment of life and travel in summer, measures to prevent haze in autumn, and the adverse effects of haze on human health in winter.

## 1. Introduction

Currently, with the continuous acceleration of China’s industrialization, there has been a significant increase in the utilization of resources and productivity. However, concomitant with economic development, multi-factor environmental and ecological problems have become increasingly prominent, directly affecting people’s normal lives and physical and mental health [1]. With growing public awareness, the deteriorating environment in China has been widely discussed [2]. For example, in December 2016, many places in China experienced prolonged high-intensity haze weather, which paralyzed flights and highways and greatly affected people’s daily lives. Living green and healthy lives has become a key social topic in China. The haze problem highlights the public’s perception and response to the environment, and whether the problem can be solved effectively depends in part on public perception and response. It is key to the success of haze control that the public is informed of the policies aimed at haze control and adjust their behavior toward reducing negative impacts. Therefore, policy-makers should actively guide the public to participate in the decision-making of haze control in real time.

In recent years, alongside continuous development, the Internet has become an indispensable part of people’s lives. Additionally, the rapid development of social media and the rapid adoption of smart phones have together provided an increasingly popular platform for people to voice their opinions [3]. The large volume of data generated on social media provides us with new means to quantify public perception or to understand the characteristics of social behavior, thereby linking the perceptions of the physical world and the psychological world. Socially-perceived computing has become an intense area of research, beginning with the seminal article, “Socially Aware Computation and Communication”, published by Alex Pentland [4]. In this article, Pantland introduced the concept of social perception computing, quantified human behavioral rhythms, movements, behavioral postures, etc., regarding the process of interpersonal communications in society, and visualized these observations to interpret human behavior rules and reveal patterns in human social interactions. Based on the concept of social perception computing, a growing number of scholars are applying new methods, such as data mining, network analysis, and machine learning, to study behavioral characteristics of the public. For example, Farrrahi [5] employed the Latent Dirichlet Allocation (LDA) and Author Topic Model (ATM) text analysis methods to explore the characteristics of individual lives and work behaviors based on data collected from mobile phones.

Weibo (or microblog) is a social media platform in China and functions like Twitter. It has already made a tremendous impact on social life and has become one of the main media platforms in China due to its rapid development, strong interactions, and easy information updates. With the rise of the web2.0 era, Weibo has experienced a rapid expansion [6]. Although data from social media such as Weibo may not be completely authentic and objective and some comments do not necessarily reflect people’s actual views and behavior [7], they can indicate people’s emotions, sentiments [8], and values. Furthermore, the rich text resources on Webio contain key data and information and, compared with other text resources, have the characteristics of high relevance and strong real-time performance [9]. These two characteristics make Weibo reflective of history, the current situation, and trends of perceived changes in the future. Note that given the limitations, it may not be used as the primary decision-making basis for air pollution control [10] but can, as argued, provide valuable information conducive to policy-making. Weibo has become a commonly used resource for social perception computing. Such methods as keyword-based word frequency, co-occurrence, and other text mining have been used to analyze changes in the public’s perception of haze weather using text information resources on Weibo. In text resources, some key and frequently used words reflect public narrative, the degree of concern, and expectations for future development [11]. Based on this, this paper used keyword semantic association analysis to construct a social perception model [12] for haze perception and carried out text data mining to study how the public perception of haze changed over time. The goals of the study were to gain a better understanding of public perception of haze weather in China and to broaden the current research field of social perception computing.

## 2. Materials and Methods

The proposed approach began with selecting comment data published by Weibo users as the main data source, and then used the methods of keyword extraction and semantic analysis data mining to effectively extract relevant key information regarding haze. Next, we visualized key information to show trends in the public’s perception of haze. The specific steps were as follows:
(1)Model construction: An analytical model on the trend of public perception of haze was constructed based on social perception calculation;(2)Data acquisition: We used “haze” and “PM2.5” as keywords and utilized crawlers to obtain microblog text data published by Weibo users from 1 January 2018 to 31 December 2018;(3)Data preprocessing: After the web crawlers obtained the microblog text resource information, the data was further cleaned by deduplication, removal of advertisement information, and processing of redundant information;(4)Keyword extraction: Natural language processing was used to extract perceptive keywords and semantic co-occurrence perception keywords from the pre-processed text data;(5)Data visualization: The extracted perceptual keywords and semantic co-occurrence perception keywords were visualized to generate a corresponding perceptual keyword co-occurrence matrix, s perceptual keyword cloud map, and a perceptual semantic network map;(6)Semantic analysis: The perceptive semantic network graph obtained through data visualization was analyzed using complex network topology characteristics;(7)Social perception analysis: Through the data mining analysis of the text information, the trend of the public’s perception of haze was further explored.


## 3. Results

In recent years, the frequent occurrence of haze across the country has caused a great threat to the physical and mental health of the public, as well as their enjoyment of life. As a result, the public responded with active discussions of the issue and formed their opinions [13,14]. Obtaining these discussions and opinions in a timely manner could help policy-makers to better understand the responses of the public, promote timely participation of policy-makers, and eventually lead to better policy-making regarding haze control (e.g., release of timely alerts and information about preventive measures).

### 3.1. Data Source

Using web crawlers, we searched the original Weibo published by Weibo users from 1 January 2018 to 31 December 2018 based on two keywords, namely, “haze” and “PM2.5”, and saved the crawled blog data to a local server as our main data source. Then, we cleaned the data by deleting duplicate content and advertisement. In addition, we divided the data for the entire whole year into four seasons, i.e., spring (March, April, and May), summer (June, July, and August), autumn (September, October, and November), and winter (December, January, and February). The trend of the public’s environmental perception of haze weather was studied on the basis of the four seasons. The number of microblog posts in different seasons is shown in Figure 1.

As shown in Figure 1, the number of micoblog posts about haze in summer declined significantly from spring, and then rose again in autumn. The upward trend from summer to autumn was quite significant, followed by a further rise until the number of posts reached the highest in winter. The change in the number of micoblog posts reflected a change in the degree of public perception of haze. Summer is the foggy season, and public perception is relatively weak, while winter is generally the high season of haze, and the public response becomes stronger.

### 3.2. Data Analysis

Although not all microblog comments are objective or authentic, an issue that is true of any social media data, microblog comment data adequately captures the emotions, feelings, and sentiments of the public. Therefore, through mining and analyzing the descriptive perception information of the public in microblog comments, it was possible to extract the most intuitive feelings and cognitive responses of the public in regard to haze weather. The high-frequency words highlighted in the Weibo comments reflected the hotspots and focus points of public attention, while the keyword co-occurrence network maps reflected the relationships between hotspots, which visually depicted the theme of public attention. This paper used the microblog data related to haze and PM2.5 to extract the keywords related to haze and PM2.5 in the blog posts and visualized the co-occurrence network map of the perceived keywords, so as to analyze the public’s environmental perception during haze weather.

#### 3.2.1. Perceptual Keyword Analysis

Firstly, after the data obtained by the reptiles was cleaned, they were divided into four seasons of spring, summer, autumn, and winter. Then, the data of the four seasons were processed by word segmentation, and keywords were extracted and treated as stop words. Next, the perceptual keywords related to the public responses to haze were obtained, and the perceived keyword frequency was also obtained. After extracting the perceived keyword frequency from the perceptual keyword, with the two known high-frequency keywords of haze and PM2.5 removed, the top 100 keywords were ranked to generate perceptive word cloud maps for all four seasons (Figure 2). In the word cloud diagram, the size of a perceived keyword represents its frequency. The larger the font displayed in the word cloud, the more popular the keywords were among the public.

It can be seen from Figure 2A that under the influence of haze in spring, keywords such as cause, composition, root cause, and pollutant composition appeared frequently, reflecting that in spring, the main concern of the public was the cause of haze.

Figure 2B reflects the obvious highlights of enjoyment in life, with keywords such as life, travel, happy play, pleasant journey, and sunny weather appearing frequently. Summer is a not a heavy haze season, so the public was less affected by haze and attention was more on enjoying life.

As can be seen from Figure 2C, in autumn, high-frequency perceived keywords were concentrated on curative methods, measures taken, scientific flood control, and filtered air. This showed that with the occurrence of haze in autumn, the public’s perception was more focused on haze prevention measures.

Figure 2D reflects the prominent frequency of keywords such as health risks, discomfort symptoms, pharyngitis, sore throat, and chest tightness. Winter is generally the high season of haze, and the public generally have a series of uncomfortable physiological reactions due to the influence of haze. It was clear that in winter the public was greatly affected by haze and had a strong reaction.

As can be seen from the four-season cloud map of perception keywords shown in Figure 2, under the influence of haze weather, the public’s perception changed depending partly on how severe the haze weather was, and the focus of attention differed between different seasons.

#### 3.2.2. Perceptual Co-occurrence Network Analysis

When two perceptual keywords appeared together in the microblog data, we described them as co-occurring and then explored the correlation strength relationship between them. Drawing a perceptual co-occurrence network map analyzed the relationship between public perception hotspots [15]. In the social network, the degree of a node reflected the importance of the node’s status in the whole network [16]. The higher the degree of a node, the higher the importance of the node status. Every node in the social network was a perceptual keyword. The degree of a node was defined as follows:
(1)$$K_{i}=\sum_{j}a_{ij}=\sum_{i=1}a_{ij}$$
where $K_{i}$ represents the degree of the *i*-node, indicating the number of nodes directly connected with other nodes. This feature easily distinguishes the particularity of the node and its impact on the entire network [17].

The connection between two keywords represented their co-occurrence relationship. The closer the connection was, the more times the two keywords co-occurred with one another. The higher the co-occurrence frequency, the stronger the correlation between words, and the greater the edge weight between nodes. To characterize the local information of a node, we introduced node strength, which was defined as:
(2)$$S_{i}=\sum_{j\in N_{i}}w_{ij}$$
where $N_{i}$ represents the set of neighbor nodes of node *i* and $S_{i}$ represents the comprehensive characteristics of the node degree k and the $w_{ij}$ [18].

In the social network, the clustering coefficient of nodes indicated the possibility that nodes would become neighbors with each other and described the closeness of the network. If the clustering coefficient was higher, the closeness between adjacent nodes was greater, that is, the degree of correlation between neighboring hotspots was stronger and the clustering coefficient clearly reflected the public’s comprehensive attention regarding the whole perceptual network. According to the definition of Barrat et al [19],
(3)$$C_{i}^{w}=\frac{\sum_{j,h}\frac{W_{ij}+W_{ih}}{2}a_{ij}a_{ih}a_{jh}}{S_{i}\left(K_{i}-1\right)}$$
where $a_{ij}$, $a_{ih}$, and $a_{jh}$ are the connection relationships of nodes *i*, *j*, *k*, and $C_{i}^{w}\in \left[0,1\right]$. We defined the average clustering $C_{i}^{w}$ coefficient of the entire network which reflected the topological structure of the entire social network and the basic characteristics of the weight distribution. The definition was as follows:
(4)$$C^{w}=\frac{\sum_{i=1,2,\cdots n}C_{i}^{w}}{n}$$
where *n* is the total number of nodes in the entire social network.

In order to further explore the trend of the public’s perception and how it changed with the seasons during haze weather, we designed a co-occurrence matrix representing the perceptual keywords and generated a perceptual keyword co-occurrence network map for each season, as shown in Figure 3, Figure 4, Figure 5 and Figure 6. Each network node represented a perceptual keyword. The size of the node represented the degree of the node in the network where the node was located, that is, the degree of co-occurrence of the node with other nodes (keywords). The larger the node, the higher the degree of co-occurrence.

To reflect the closeness of each network as a whole, we first obtained the network average clustering coefficient of the four seasons according to Equation (4), and then counted the clustering coefficients of all four collinear network node keywords according to Equations (1)–(3), and ranked the top ten keywords with clustering coefficients, as shown in Table 1.

From Figure 3, we can see that haze and PM2.5 had the largest nodes in the co-occurrence network map in spring, that is, the node degree was the highest, indicating that the two were at the core position in the entire shared network. At the same time, this unsurprisingly reflected that haze and PM2.5 were the core hotspots of public concern in haze weather. In addition, we can see from Table 1 that the top-ranked perceptual keywords with the clustering coefficient included generation, composition, root cause, pollutant composition, formation, structure, tectonic attribution, pollution source, pollution attribution, and air pollution. The formula $C_{i}^{w}$ > $C^{w}$ showed that they were highly concentrated and had strong correlations with haze and PM2.5 at the core position, i.e., these sensory keyword node clusters were also hotspots of public concern. The relationships between these hotspots and perceived keywords reflected the public’s concern about the causes and composition of haze under the influence of haze weather in spring. In addition, there were some perceived keywords at the edge of the network, such as escape, mood, confusion, endurance, trouble, nervousness, and unhappiness. These perceived keywords had a relatively low degree of centrality and their connections with other core keywords were relatively sparse, indicating that these words were “emotional” hotspots generated by the public’s attention to the causes of haze in haze weather.

Figure 4 shows that haze and PM2.5 were still the most important perceptual keywords with the highest degree of nodes in the summer co-occurrence network map. While haze and PM2.5 were still the core hotspots of the public, at the same time it can be seen from Table 1 that other keywords, such as enjoying life, travel, happy play, pleasant roads, beautiful winds, disappearing days, hiking, sunny, and high-quality air had higher clustering coefficients, that is, the degree of aggregation was higher, and $C_{i}^{w}$ > $C^{w}$. These hotspot keyword node clusters reflected that during the summer, the public’s attention turned to travel and other activities related to enjoyment in life. At the edge of the network, there were some positive emotions, such as laughter, happiness, openness, carnival, happiness, and freedom. Since the summer was not a hazy season, the public’s satisfaction with life also increased.

In Figure 5, haze and PM2.5 were still the central perceptual keywords in the autumn co-occurrence network map. It can be seen from Table 1 that the key factors in the clustering coefficient ranking were cure method, measures taken, scientific flood control, boycott of the sky, flood control measures, keeping clean, masks, filtering the air, scientific prevention, and protection. These were the keywords to prevent haze,$C_{i}^{w}$ > $C^{w}$; these perception keywords had relatively high clustering coefficients, a large degree of aggregation, and a strong correlation with the core perceptual keywords, reflecting that the public was most concerned about the haze prevention measures under autumn haze. At the edge of the network, there were some negative emotional perception keywords that accompanied haze, such as nervousness, alertness, sadness, and disconsolate.

In Figure 6, haze and PM2.5 were still the most core perceptual keywords in winter. It can be seen from Table 1 that the clustering coefficient ranked more positive than the perceived key words, including health hazards, discomfort, pharyngitis, sore throat, phlegm, cough, asthma attack, chest tightness, ecological crisis, and chronic disease, indicating that these perceptual keywords were more concentrated, and $C_{i}^{w}$ > $C^{w}$. These were perceived keywords that showed symptoms of physical discomfort. Haze has the most direct health impact on the public and commonly produces uncomfortable physiological reactions as well as a kind of feedback on the impact of haze on people’s travel. At the edge of the co-occurrence network, there were some relatively negative emotional perception keywords like unhappy, depressed, sad, bad, blocked, lost, pessimistic, and dull, meaning that haze triggered the public’s negative emotions.

## 4. Discussion

Due to the obvious seasonality of haze, we also applied socially-aware analysis methods. It could be seen from the social perception network co-occurrence map that under the influence of haze weather, the public’s perception intention changed with the seasons.

From Figure 3 combined with Table 1, we can see that under the influence of the haze weather in spring, the perception was concentrated on the causes of haze and the composition of haze. The main reason for this is that spring is the starting season of the year and is also the season when haze begins to appear. Compared with more severe haze seasons, the degree of haze pollution in spring is relatively mild, and public attention focuses on the causes of haze.

It can be seen from Figure 4 and Table 1 that the end of spring brought the season with the least haze pollution in a year, and the public was also least affected by haze, so the public showed a more positive response and mood. They spend the summer on traveling, enjoying oneself, and other activities. The growth of vegetation is vigorous in summer, which also plays a role in air purification and contributes to the public’s positive emotions.

When summer was over and fall was coming, the public expected that the arrival of autumn would be accompanied by haze. Due to the public’s understanding of haze as a threat to human health, we can see from Figure 5 combined with Table 1 that the public perception was more focused on the preventive measures of haze.

In winter, due to low temperatures, the atmospheric layer is relatively stable and wind speeds are low. These factors, together with increased humidity, cause the stagnation of polluted particles and haze. Because haze causes direct discomfort to the human body, especially to the respiratory system, the public’s perception focused on their own body discomfort symptoms, as we can see from Figure 6 and Table 1. In addition, haze caused social security problems, such as traffic accidents and low visibility.

## 5. Conclusions

This paper uses the public’s microblog comments from the Internet as basic data to analyze the public’s perception of environmental quality changes. Taking haze weather in China as a case study, this paper applies the network topology analysis method to analyze the change in public perception of the environment, with the aim of demonstrating to the environmental management departments that the public’s environmental mood can be captured in a timely manner, thus providing a decision-making basis for them to formulate corresponding pollution prevention measures. This study has perspective applications for public participation in environmental pollution supervision.

Through our research, it was found that the public’s environmental perception with respect to haze weather had a significant seasonality. In spring, the public’s environmental perception was focused on the causes and composition of haze. Similar to macroscopic causes, such as the incineration of garbage, industrial concentration, coal burning, automobile exhaust, increased private cars, and exhaust emissions, there are also microscopic causes that contribute to haze formation, including particulate matter, nitrogen oxides, aerosols, and sulfates. In summer, the public’s perception was mainly focused on visually sensitive weather and environmental features, such as sunny, green mountains, haze dissipation, high-quality air, suddenly bright, hot summer, full of happiness, pleasing to the eye, close to nature, and other sunny weather. In autumn, the public’s perception of environment tended to focus on the preventive measures against haze and relevant self-protection behaviors after haze becomes more severe and visible. During this time, high-frequency perceptual keywords included anti-mite measures, masks, scientific prevention, clean, green energy, air purification, and increased green plants. In winter, with the arrival of the high season of haze, the public directly felt the severe impact of haze weather, which especially affected the respiratory system. Hence, the public’s perception was mainly focused on the physical symptoms during haze weather, such as cough, skin allergies, phlegm, headache, pharyngitis, lung discomfort, chronic diseases, allergic rhinitis, and chest tightness, as well as adverse social problems caused by haze, such as traffic jams, traffic accidents, flight delays, suspension of classes, gray, and visibility.

This study explored how the public’s perception of haze during haze weather changed with the seasons. Most studies carried out in this are used surveys or interviews, with few using data from social medial such as microblog (or Weibo). Compared with other methods, Weibo comment data provides a more effective means of extracting relevant information from the public. Admittedly, the information extracted from Weibo alone may be limited and not fully representative of what the public thinks, as there is still a large proportion of the population that does not use Weibo. Future studies could explore other social media platforms or pursue a multi-platform approach to more comprehensively capture the public’s responses to haze weather and other environmental issues. One focus of future studies may be on how the Chinese public perceive climate change and what actions they take to reduce emissions and adapt to the changing climate, considering that climate change impacts are already taking place and are projected to especially affect developing countries.

## Figures and Tables

**Figure 1 ijerph-16-04767-f001:**
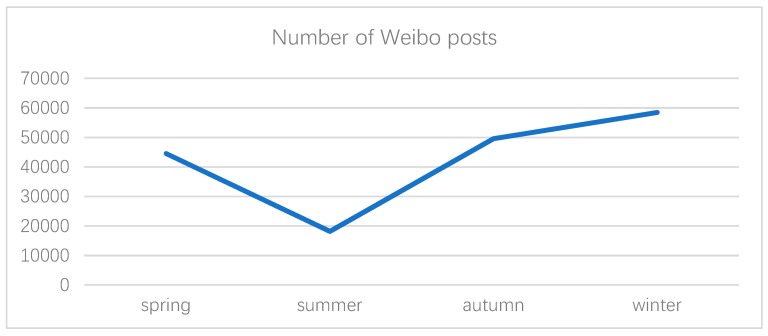
The number of Weibo posts varied from season to season.

**Figure 2 ijerph-16-04767-f002:**
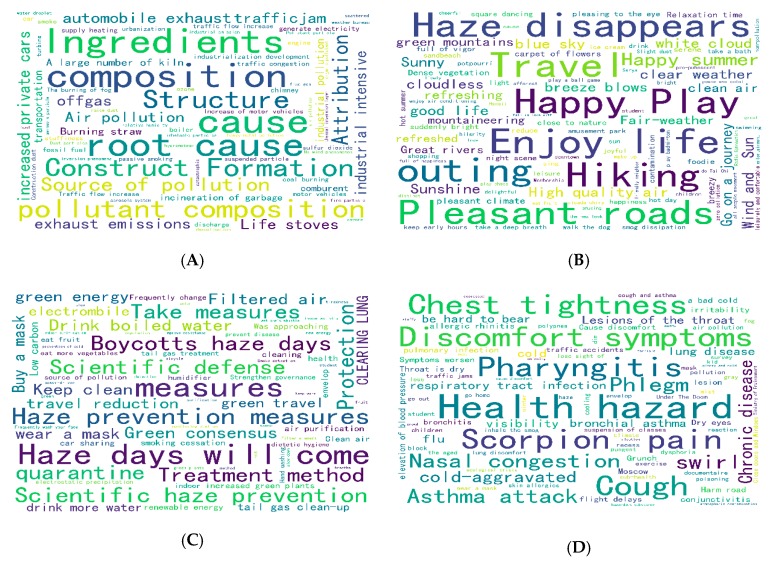
(**A**) Spring micro-blog perception keywords cloud map. (**B**) Summer micro-blog perception keywords cloud map. (**C**) Autumn micro-blog perception keywords cloud map. (**D**) Winter micro-blog perception keywords cloud map.

**Figure 3 ijerph-16-04767-f003:**
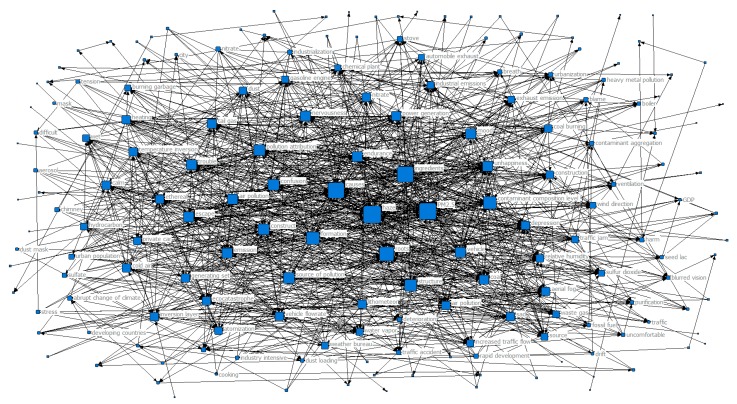
Network map of keyword co-occurrence in spring.

**Figure 4 ijerph-16-04767-f004:**
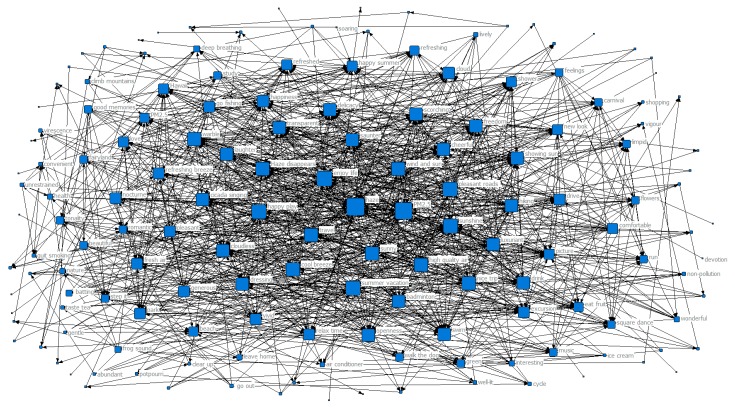
Network map of keyword co-occurrence in summer.

**Figure 5 ijerph-16-04767-f005:**
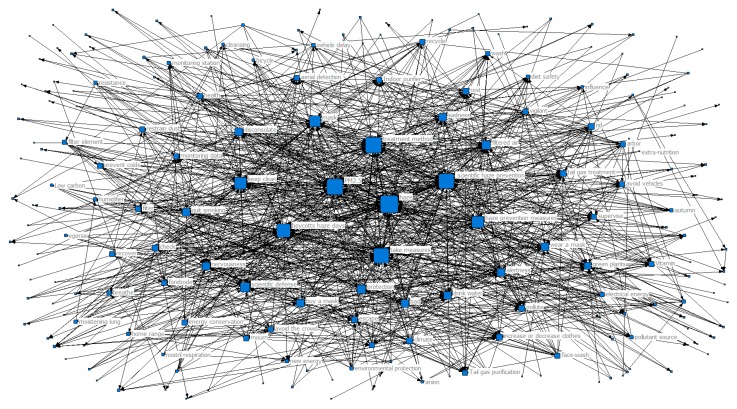
Network map of keyword co-occurrence in autumn.

**Figure 6 ijerph-16-04767-f006:**
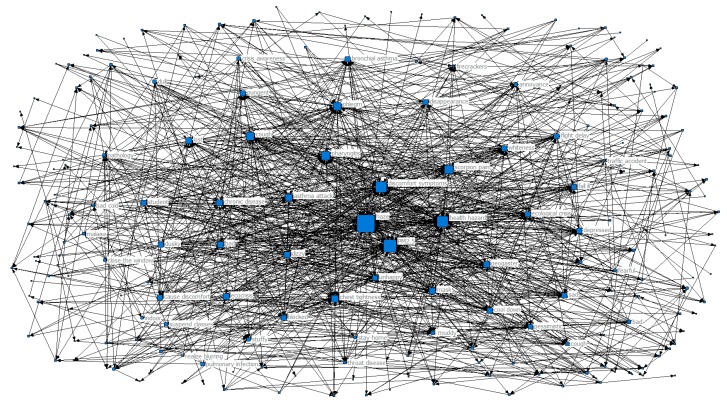
Network map of keyword co-occurrence in winter.

**Table 1 ijerph-16-04767-t001:** Top ten keyword co-occurrence network clustering coefficients over the four seasons.

Season	Spring	Summer	Autumn	Winter
$C^{w}$	0.598	0.588	0.601	0.612
1	Causes 0.685	Enjoy Life 0.692	Treatment Method 0.715	Health Hazard 0.727
2	Ingredients 0.681	Travel 0.678	Take Measures 0.710	Discomfort Symptoms 0.723
3	Roots 0.674	Happy Play 0.671	Scientific Haze Prevention 0.709	Pharyngitis 0.716
4	Contaminant Composition Level 0.675	Pleasant Roads 0.668	Boycotts Haze Days 0.694	Scorpion Pain 0.703
5	Formation 0.663	Wind And Sun 0.662	Haze Prevention Measures 0.687	Phlegm 0.695
6	Structure 0.652	Haze Disappears 0.647	Keep Clean 0.675	Cough 0.684
7	Construct 0.648	Hiking 0.639	Mask 0.658	Asthma Attack 0.661
8	Source of Pollution 0.637	Sunshine 0.634	Filtered Air 0.631	Chest Tightness 0.652
9	Pollution Attribution 0.626	Sunny 0.621	Scientific Defense 0.618	Ecological Crisis 0.647
10	Air Pollution 0.619	High Quality Air 0.607	Protection 0.611	Chronic Disease 0.634

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
