# Peer review of "Public Perception of Haze Weather Based on Weibo Comments"

_ijerph, 2019, doi:10.3390/ijerph16234767_

Round 1
Reviewer 1 Report
Table 1, why the value in autumn for Scientific haze prevention is 10.709? What does this value imply? The authors need to discuss in the revised manuscript. The creativity of this study need to be stated in the revised manuscript. What are the perspective scientific contributions in the predictions or preventions of PM2.5? Can this method also apply to the other pollutants species such as heavy metals, ionic species which attached on the PM2.5? The authors need to ask for a native English professor to help edit the English in the revised manuscript. It should be include the whole text and the grammar. This manuscript can be accepted for published in this journal after the above major comments have been addressed.Author Response
Point 1: Table 1, why the value in autumn for Scientific haze prevention is 10.709? What does this value imply? The authors need to discuss in the revised manuscript.
Response 1: Thanks for pointing it out. This is a mistake. The correct value should be 0.709.
Point 2: The creativity of this study need to be stated in the revised manuscript. What are the perspective scientific contributions in the predictions or preventions of PM2.5? Can this method also apply to the other pollutants species such as heavy metals, ionic species which attached on the PM2.5?
Response 2: This paper uses the public's microblog comments from the Internet as the basic data to analyze the public's perception of environmental quality changes. Taking haze weather in China as a case study, this paper applies the network topology analysis method to analyze the change in the public’s environment perception, with the aim i) of demonstrating to the environmental management departments that the public’s environmental mood can be captured in a timely manner and thus ii) of providing a decision-making basis for them to formulate corresponding pollution prevention measures. Since PM2.5 is currently the primary pollutant of air quality in China, and is widely discussed among the public, PM2.5 is selected as the research object. Certainly, the proposed method is also suitable for other pollutants in other regions. We added text to highlight the novelty of the paper in Conclusions. Please see lines 291-297.
Point 3: The authors need to ask for a native English professor to help edit the English in the revised manuscript. It should be include the whole text and the grammar. This manuscript can be accepted for published in this journal after the above major comments have been addressed.
Response 3: Thanks for the suggestion. The manuscript has been edited by a professional English-language editing company. If needed, we can provide the invoice from the company.

Reviewer 2 Report
Referee report -Public Environmental Perception Based on Weibo Comments in Haze Weather
This is an excellent paper which provides important and innovative policy advice. The goals of managing and formulating effective environmental policies are clearly and explicitly stated. The more general objective is obviously to improve the quality of life through a set of easily applicable IT measures. Social public environment perceptions and public concerns about the natural environment are a highly important, although extraordinarily complex, topic of research, so that some adjustments is required (see below)
I have therefore three comments and suggestions for change:
First, the article is pervaded by an unwarranted technological optimism, conveyed by the certainty that the Internet may offer the ideal instrument required for action and urgent change. Even though the article is reasonably successful in achieving its goals, it is imperative that some of the most unnecessary extra-confidence is nuanced, so that one doesn't have the impression that social networking can offer a sort of magic wand. This can be easily achieved thorough minor changes, including comments in the text or in the footnotes about the well known limits of the Internet. References to work which addresses this problem from various angles include 'Irresponsible radicalisation: Diasporas, globalisation and long-distance nationalism in the digital age', JEMS, 2012 38, 9, pp. 1357-1379, 2012.
As a second point, the problem of pollution appears here to be a very local one which can be easily resolved through adopting appropriate measures, including those indicated. This is, however, a weak point in the article. The authors do not mention the broader problems of pollution exceeding city limits and even national boundaries. In particular, there is no mention of climate change, as if private car pollution can be abstracted from the broader problem of its global environmental impact which affect not only the local habitats but the rest of the world.
Climate change emergency is obviously a broader and more comprehensive problem than can be barely addressed by mere environmental action, although local measures are very important. The authors need to be clear about that.
In other words, I am not sure, the authors sufficiently contextualise the problem of haze within a broader non-local perspective. For instance, does the public's environmental awareness of the haze weather include such broader dimensions? Does social perception change accordingly? One has the impression that the urgency of the climate change is not fully appreciated.
Please also bring in statistics about the destructive impact of SUVs on both haze and climate.
Thirdly, the article concludes by showing how public environmental perceptions change with the change of seasons. Even through this assessment is limited to the case of haze, one is surprised to see no reference to the usefulness of more traditional forms of knowledge. Since we are also talking about the importance of spreading knowledge about current policies (considering the enormous size of the problem), it may be necessary to mention the need to incorporate notions of traditional knowledge.
There are articles on this topic which the authors should consider in order to contribute a more significant and enriching argument.
Some of them have been published in this journal and I think they could be included even though they my not seem relevant at first glance to the Chinese case. They include: 'Bi-Directional Learning: Identifying Contaminants on the Yurok Indian Reservation ' or 'Assessing the Impacts of Local Knowledge and Technology on Climate Change Vulnerability in Remote Communities'.
This shows how new technologies can alter the way individuals acquire and transmit knowledge about their surrounding environment. Some of these technologies further increase the distance between individuals and their environment. Thus the transfer of traditional local knowledge is also an essential ingredient, particularly essential is the inter-generational transmission considering the tensions between technology-induced distancing and local knowledge exerted on community vulnerability. Haze perceptions could also be influenced by the movement away from traditional means of transport. In this context, the transmission of knowledge mitigates the potentially adverse effects of technology-induced distancing. So knowledge transfer is also essential in assessing perceptions of how the haze phenomenon has varied across town.
On this, see also:
‘Knowledge Transfer and Exchange Processes for Environmental Health Issues in Canadian Aboriginal Communities’ (also in this journal). For a more general approach see Levene's important contribution: 'Subsistence societies, globalisation, climate change and genocide: discourses of vulnerability and resilience', IJHR, 18, 3, pp. 281-297.
As a general theoretical framework, I suggest taking a look at this powerful argumentation on the global ramification of the issues raised above.
One more questions: What exactly is meant by 'method of interdisciplinary and integration' (p. 1)?
Finally, I find it surprising that the authors do not mention similar research published elsewhere. For instance, here is a further suggestion for expending the poor references apparatus: 'Can Social Media Clear the Air? A Case Study of the Air Pollution Problem in Chinese Cities', The Professional Geographer, 67, 3, pp. 351-363. 2015. Could you please also state at some point what innovation could you offer in respect to this paper?
In general, the bibliography is poor and faulty and needs to be enriched with more references.
Author Response
Referee report- Public Environmental Perception Based on Weibo Comments in Haze Weather This is an excellent paper which provides important and innovative policy advice. The goals of managing and formulating effective environmental policies are clearly and explicitly stated. The more general objective is obviously to improve the quality of life through a set of easily applicable IT measures. Social public environment perceptions and public concerns about the natural environment are a highly important, although extraordinarily complex, topic of research, so that some adjustments is required (see below)
Thanks for the positive assessment of our study.
Point 1: First, the article is pervaded by an unwarranted technological optimism, conveyed by the certainty that the Internet may offer the ideal instrument required for action and urgent change. Even though the article is reasonably successful in achieving its goals, it is imperative that some of the most unnecessary extra-confidence is nuanced, so that one doesn't have the impression that social networking can offer a sort of magic wand. This can be easily achieved thorough minor changes, including comments in the text or in the footnotes about the well known limits of the Internet. References to work which addresses this problem from various angles include 'Irresponsible radicalisation: Diasporas, globalisation and long-distance nationalism in the digital age', JEMS, 2012 38, 9, pp. 1357-1379, 2012.
Response 1: Thanks for the suggestion. We added more text to describe the limitations of using data from social media platforms. Please see lines 56-58 and 320-322.
Point 2: As a second point, the problem of pollution appears here to be a very local one which can be easily resolved through adopting appropriate measures, including those indicated. This is, however, a weak point in the article. The authors do not mention the broader problems of pollution exceeding city limits and even national boundaries. In particular, there is no mention of climate change, as if private car pollution can be abstracted from the broader problem of its global environmental impact which affect not only the local habitats but the rest of the world. Climate change emergency is obviously a broader and more comprehensive problem than can be barely addressed by mere environmental action, although local measures are very important. The authors need to be clear about that.In other words, I am not sure, the authors sufficiently contextualise the problem of haze within a broader non-local perspective. For instance, does the public's environmental awareness of the haze weather include such broader dimensions? Does social perception change accordingly? One has the impression that the urgency of the climate change is not fully appreciated.Please also bring in statistics about the destructive impact of SUVs on both haze and climate.
Response 2: Thanks for the questions. Our paper focuses on haze, which is perhaps the top public concern in China. Through the analysis of changes in public perception, our paper aims to provide a decision-making basis for improving the system design of public policy to improve people’s well-being. We understand that haze is mostly a local and regional environmental problem. Global problems such as climate change are beyond the scope of this study, so our results may not be applicable to climate change. We agree with the reviewer, though, that it may be interesting to study climate change. In response to your comments, we added some text in the end to suggest that future research may explore how the public perceive climate change in China. Please see lines 364-367.
Point 3: Thirdly, the article concludes by showing how public environmental perceptions change with the change of seasons. Even through this assessment is limited to the case of haze, one is surprised to see no reference to the usefulness of more traditional forms of knowledge. Since we are also talking about the importance of spreading knowledge about current policies (considering the enormous size of the problem), it may be necessary to mention the need to incorporate notions of traditional knowledge.
There are articles on this topic which the authors should consider in order to contribute a more significant and enriching argument.Some of them have been published in this journal and I think they could be included even though they my not seem relevant at first glance to the Chinese case. They include: 'Bi-Directional Learning: Identifying Contaminants on the Yurok Indian Reservation ' or 'Assessing the Impacts of Local Knowledge and Technology on Climate Change Vulnerability in Remote Communities'.This shows how new technologies can alter the way individuals acquire and transmit knowledge about their surrounding environment. Some of these technologies further increase the distance between individuals and their environment. Thus the transfer of traditional local knowledge is also an essential ingredient, particularly essential is the inter-generational transmission considering the tensions between technology-induced distancing and local knowledge exerted on community vulnerability. Haze perceptions could also be influenced by the movement away from traditional means of transport. In this context, the transmission of knowledge mitigates the potentially adverse effects of technology-induced distancing. So knowledge transfer is also essential in assessing perceptions of how the haze phenomenon has varied across town.
On this, see also:‘Knowledge Transfer and Exchange Processes for Environmental Health Issues in Canadian Aboriginal Communities’ (also in this journal). For a more general approach see Levene's important contribution: 'Subsistence societies, globalisation, climate change and genocide: discourses of vulnerability and resilience', IJHR, 18, 3, pp. 281-297.As a general theoretical framework, I suggest taking a look at this powerful argumentation on the global ramification of the issues raised above. One more questions: What exactly is meant by 'method of interdisciplinary and integration' (p. 1)?.
Response 3: Thanks for the questions. Our paper focuses on key information revealed on social media as representation of the public’s perception. It does not examine exactly how people respond or what preventative measures they take and where they obtain their knowledge (traditional or from online or from school). What you suggested is beyond the scope of this paper. But thanks for the suggestion, nevertheless. Our future research will consider this aspect.
Point 4: Finally, I find it surprising that the authors do not mention similar research published elsewhere. For instance, here is a further suggestion for expending the poor references apparatus: 'Can Social Media Clear the Air? A Case Study of the Air Pollution Problem in Chinese Cities', The Professional Geographer, 67, 3, pp. 351-363. 2015. Could you please also state at some point what innovation could you offer in respect to this paper?In general, the bibliography is poor and faulty and needs to be enriched with more references.
Response 4: Thanks for the suggestion and questions. We have cited the reference you kindly suggested (Please see lines 59) and also added some other related references. The novelty of our paper is added. This paper also argues that social media cannot be fully used as the main basis and means of air pollution control, but the public's perception and satisfaction with respect to environmental pollution can be obtained based on analysis of social media data, and can be used as a reference for government management departments to make relevant policies.

Round 2
Reviewer 1 Report
It is an informative and interesting study. However, i sugges the authors should ask for a native English professor or speaker to help edit the whole article in the revised manuscript. It should include the grammar and the text. What are the industrial and perspective applications of this study? The authors should disxuss it in the revised manuscript. This manuscript can be accepted for published in this journal after the above minor comments have been addressed.Author Response
Point 1: It is an informative and interesting study. However, i suggest the authors should ask for a native English professor or speaker to help edit the whole article in the revised manuscript. It should include the grammar and the text.
Response 1: The manuscript has been edited by a professional English-language editing company.
Point 2: What are the industrial and perspective applications of this study? The authors should discuss it in the revised manuscript. This manuscript can be accepted for published in this journal after the above minor comments have been addressed.
Response 2: This study has perspective applications for public participation in environmental pollution supervision. Please see lines 299-300.
